# Improving the Extraction Process of Mehlich 3 Method for Calcareous Soil Nutrients

**Changqing Li** [1], **Shuo Wang** [1], **Mengyu Sun** [1], **Dongxiao Li** [2], **Huasen Xu** [1], **Liangyu Zhang** [1], **Cheng Xue** [1], **Wenqi Ma** [1] and **Zhimei Sun** [1,*]

1   Key Laboratory for Farmland Eco-Environment of Hebei, College of Resources and Environmental Sciences, Hebei Agricultural University, Baoding 071000, China
2   College of Agronomy, Hebei Agricultural University, Baoding 071000, China
*   Correspondence: sunzhimei@hebau.edu.cn; Tel.: +86-13930892021

**Abstract:** Soil nutrient testing is an effective way to uncover soil nutrient status. However, the conventional testing method (CT method) and Mehlich 3 method (M3 method) demand long-time consumption, tedious testing steps, high testing cost, dangerous chemicals contained in extractant, etc. Therefore, it is important to develop a new rapid test method or improve the existing rapid test method of soil available nutrients. In this study, an improved Mehlich3 method (IM3 method) with the new combined extractant were developed and evaluated on the testing feasibility, precision, efficiency, and cost. The results showed that: (1) IM3 method avoided the usage of two hazardous chemicals, i.e., ammonium nitrate and nitric acid, which were difficult in purchase and storage but contained in the combined extractant of M3 method. (2) The correlation coefficients of available P, K, Fe, Mn, Cu and Zn in calcareous soil between M3 and CT, and between IM3 and CT methods all reached highly significant level. The correlation coefficient of available Zn between IM3 and CT method was significantly higher than that between M3 and CT method, and those of the other elements had not obviously changed. (3) The variation coefficients of available P, K, Fe and Cu determined by M3 and IM3 methods were all lower than those determined by CT method. The variation coefficients of available Mn and Zn determined by IM3 were 3.67% and 6.43%, which were slightly higher than those determined by CT method with 2.72% and 5.29%, but were lower than those determined by M3 method. (4) Under the premise of ensuring testing precision, IM3 method took only 6.3 min/piece for determining six elements, reducing testing time by 70.7% and 3.08% compared with CT (21.5 min/piece) and M3 (6.5 min/piece) methods, respectively. The testing cost of IM3 method was reduced by 26% and 61.2% compared with M3 and CT methods, respectively. In conclusion, IM3 method is an ideal rapid measurement method for the simultaneous determination of available soil nutrients in calcareous soil.

**Keywords:** improved testing method for soil nutrients; extractant; testing precision; testing time; testing cost; calcareous soil

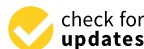



## 1. Introduction

The application of fertilizers plays an essential role in maintaining grain yield and quality. However, the long-term irrational fertilization has caused serious consequences, such as the increase of agricultural production cost, the waste of resource and energy, environment pollution, etc. [1–3]. How to improve the fertilizer utilization efficiency has become an urgent issue to ensure safe, high-quality, stable-yield, and environment-friendly agricultural production. It is widely known that soil testing and formula fertilization technology is an effective way to uncover soil nutrient contents and design fertilizer formula, which is an important contributor in order to realize the high nutrient utilization efficiency [4,5].

At present, CT method is generally used to determine soil available nutrients contents because of the precision advantage in all testing procedures. However, there still exist

some disadvantages, such as tedious testing procedures, long testing time, and high testing cost [6–9]. The combined extraction methods of ASI (systematic approach for soil nutrient status evaluation) and M3 have been widely used at home and abroad. Both methods can be used to extract various mineral elements simultaneously [10–13] with a significant correlation between both and CT method results [14,15]. However, some limitations in ASI and M3 methods have still been found. For ASI method, it is difficult to be popularized because that the determination step is complicated and special equipment is required [16,17]. For M3 method, the combined extractant contains two dangerous chemicals, i.e., ammonium nitrate and nitric acid, both are expensive and dangerous to be limited in purchase, storage and usage. All these brings inconvenience to the configuration of combined extractant and increases the testing cost. At the same time, the measurement result of M3 method is significantly higher than that of the CT method because of its lower pH [18]. Therefore, it is important to develop a new rapid test technology or improve existing rapid test technology of soil available nutrients to promote the extension of soil testing and formula fertilization technology.

Calcareous soil, one of the most typical soil types, was widely distributed in the north and northwest China. In this study, an improved M3 method (i.e., IM3 method) with the new combined extractant were developed according to analyze the merit and demerit of M3 method. Based on the hypothesis that this method is feasible, our objectives are as follows: (1) to investigate the correlation between IM3 method and CT method and the test precision of IM3 method; and (2) to compare the difference of testing time and cost between IM3 method and CT method. These results would provide technical support for the application and extension of soil testing and formula fertilization technology.

## 2. Materials and Methods

### 2.1. Soil Sampling and Characterization

The soil samples in a 0–20 cm soil layer were collected from typical farmland (52), vegetable garden (69) and orchard (61) in Hebei province, which were all calcareous soil, and located in Qingyuan, Lixian, Gaoyang, Dingzhou, Suning, Hejian and Xinji county, respectively (Figure 1). Soil samples from these sites were air-dried and passed through 2-mm sieve.

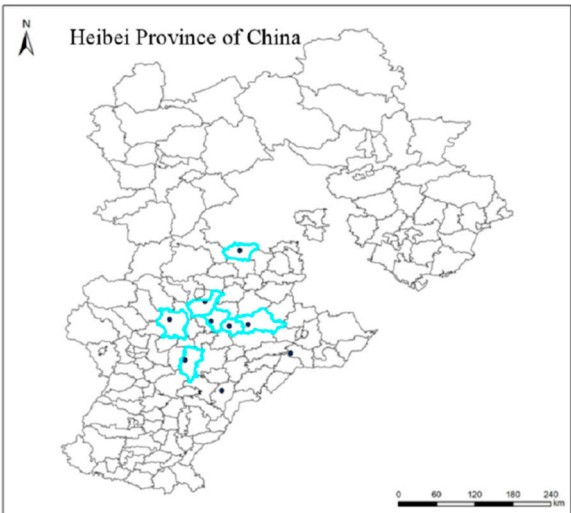

**Figure 1.** The collection areas of soil samples in Hebei province of China.

In the process of soil testing, the standard soil sample was used as a reference to clarify the accuracy of the test results. The standard sample was bought from Institute of Geophysical and Geochemical Exploration, Chinese Academy of Geological Sciences (Shijiazhuang, China).

According to WRB (2015) soil classification, the soil type was fluvisol soil, including standard soil sample and tested soil samples [19], and the basic physical and chemical properties were shown in Table 1.

**Table 1.** The basic soil physical and chemical properties of collected soil samples.

| Soil | $n$ | pH | Organic Matter (g/kg) | Available P (mg/kg) | Available K (mg/kg) | Available Fe (mg/kg) | Available Mn (mg/kg) | Available Cu (mg/kg) | Available Zn (mg/kg) |
|---|---|---|---|---|---|---|---|---|---|
| Standard sample | 1 | 8.0 | - | 36 | 217 | 50 | 25 | 1.9 | 1.8 |
| Farmland | 52 | 7.5–8.6 | 3.7–18.5 | 0.1–47.1 | 69.6–181 | 3.5–17.3 | 1.1–7.1 | 0.4–8.6 | 0.3–9.8 |
| Vegetable | 69 | 7.2–8.2 | 11.4–39.2 | 0.9–215 | 31.3–271 | 7.9–26.3 | 2.4–8.3 | 0.8–12.3 | 1.4–11.0 |
| Orchard | 61 | 7.4–8.3 | 7.3–20.7 | 3.6–322 | 56.7–660 | 8.9–28.7 | 3.3–9.0 | 1.6–12.0 | 2.5–11.4 |

*2.2. Experimental Design*

2.2.1. Improvement of Extractant in M3 Method

Taking CT method and M3 method as the control, IM3 method was established with improving the combined extractant of M3. The pH, composition of extractant, and soil-solution ratio during the extraction process by the above three test methods were shown in Table 2. The available P, available K, available Fe, Mn, Cu and Zn contents of all tested soil samples were extracted by the above three methods, respectively. The content of available P in extractant was determined by spectrophotometer (Bowei Instrument Technology Co., Ltd., Qidong, China), that of available K was determined by flame photometer (Shandong Fangke Instrument Co., Ltd., Weifang, China), and those of available Fe, Mn, Cu and Zn were determined by atomic absorption spectrophotometer (Xuzhou Chenfei Technology Development Co., Ltd., Xuzhou, China).

**Table 2.** Comparison of three test methods for determination of soil available nutrients.

| Methods | Determination | Composition of Extractant | pH | Soil:Solution |
|---|---|---|---|---|
| | Available P | $NaHCO_3$ | 8.5 | 1:20 |
| CT | Available K | $NH_4OAc$ | 7.0 | 1:10 |
| | Available Fe, Mn, Cu and Zn | DTPA, $CaCl_2$, TEA | 7.3 | 1:2 |
| M3 | Available P, K, Fe, Mn, Cu and Zn | EDTA, $NH_4F$, HOAc, $HNO_3$, $NH_4NO_3$ | 2.5 | 1:10 |
| IM3 | Available P, K, Fe, Mn, Cu and Zn | EDTA, $NH_4F$, HOAc, $NH_4Cl$ | 3.6 | 1:10 |

2.2.2. Comparison of Testing Precision of Three Methods

Five soil samples were randomly selected from the above tested soil. In addition, the contents of soil available P, available K, available Fe, Mn, Cu and Zn were determined by CT method, M3 method and IM3 method, respectively. Soil samples with eight replicates were determined with each method, and the variability of these determination results obtained by different test methods were analyzed.

2.2.3. Comparison of Testing Time and Testing Cost of Three Methods

With the exception of the preparation of extracting liquid of soil samples, the following determining procedures were the same for above three test methods. Therefore, the comparison of testing time and testing cost were only considered in the preparation stage of extracting liquid. A total of 30 samples were randomly selected from the tested soil and the contents of soil available P, available K, available Fe, Mn, Cu and Zn were determined by CT, M3 and IM3 methods, respectively. The consumed time of configuring extractant, weighing sample, adding extractant, shaking and filtrating soil samples was recorded, respectively. Finally, the average time of testing each soil sample was calculated.

For the same reason, the testing cost only considered the consumption of reagent and quantitative filter paper in the preparation stage of extracting liquid. Here, 50 soil samples were taken as a batch. The prices of reagents and quantitative filter paper bought from the

official website of Aladdin (Shanghai Aladdin Bio-Chem Technology Co., Ltd., Shanghai, China) were as follows: $NaHCO_3$: 33.4 yuan/500 g, $NH_4OAc$: 36.4 yuan/500 g, DTPA: 230.6 yuan/500 g, $CaCl_2$: 42.5 yuan/500 g, TEA: 41.5 yuan/500 mL, EDTA: 59.7 yuan/250 g, $NH_4F$: 69.8 yuan/500 g, $HNO_3$:126.5 yuan/500 mL, $NH_4NO_3$: 222.5 yuan/500 g, HOAc: 39.4 yuan/500 mL, $NH_4Cl$: 29.3 yuan/500 g, and quantitative filter paper (diameter: 150 mm): 33 yuan/100 sheets. The reagents all had a purity of AR (Analytical Reagent) grade.

### 2.3. Statistical Analysis

Statistical analysis was performed by one-way ANOVA processes of SPSS 22 software (v22.0, Chicago, IL, USA). A significant difference was determined as $p < 0.05$ using Duncan's multiple range test. The linear fitting was performed with the results measured by M3, IM3 and CT methods.

## 3. Results

### 3.1. Correlation Analysis of Three Test Methods

There was a high significantly correlation between the contents of soil available P (or K) measured by M3 and CT methods; and the same result was observed between IM3 and CT methods. The correlation coefficients $R^2$ were 0.9292 (M3-P with CT-P), 0.9031 (IM3-P with CT-P), 0.9584 (M3-K with CT-K) and 0.9244 (IM3-K with CT-K), respectively (Figure 2). The correlation coefficients of IM3 method and CT method were not significant difference from that of M3 method and CT method. The regression equations were y = 2.0533x + 2.4518 (M3-P with CT-P), y = 2.4824x + 46.219 (IM3-P with CT-P), y = 1.0077x + 4.6566 (M3-K with CT-K) and y = 0.8492x + 15.904 (IM3-K with CT-K), respectively. These results indicated that IM3 was a feasible method for extracting available P and available K in calcareous soils.

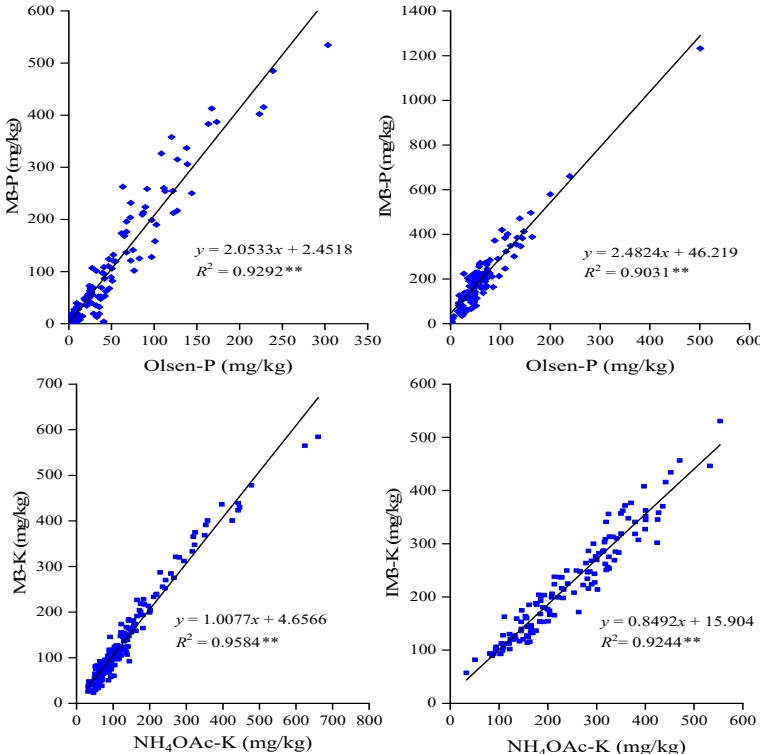

**Figure 2.** The correlations of the contents of soil available P and available K determined by three test methods (** $p < 0.01$).

The contents of available Fe, Mn, Cu and Zn determined by M3, IM3, and CT methods also showed significant correlation (Figure 3). The correlation coefficient ($R^2 = 0.9685$) of available Cu between M3 and CT was the highest, but this value of available Fe between

M3 and CT was the lowest for $R^2$ being 0.5116. The correlation analysis on IM3 and CT methods also showed the correlation coefficient of available Fe was the lowest ($R^2$ = 0.4072). The correlation coefficient of available Zn and Mn between IM3 and CT were evidently higher than that between M3 and CT, for which the correlation coefficient of Zn increased from 0.6718 to 0.9478 and that of Mn increased from 0.8179 to 0.8534, respectively.

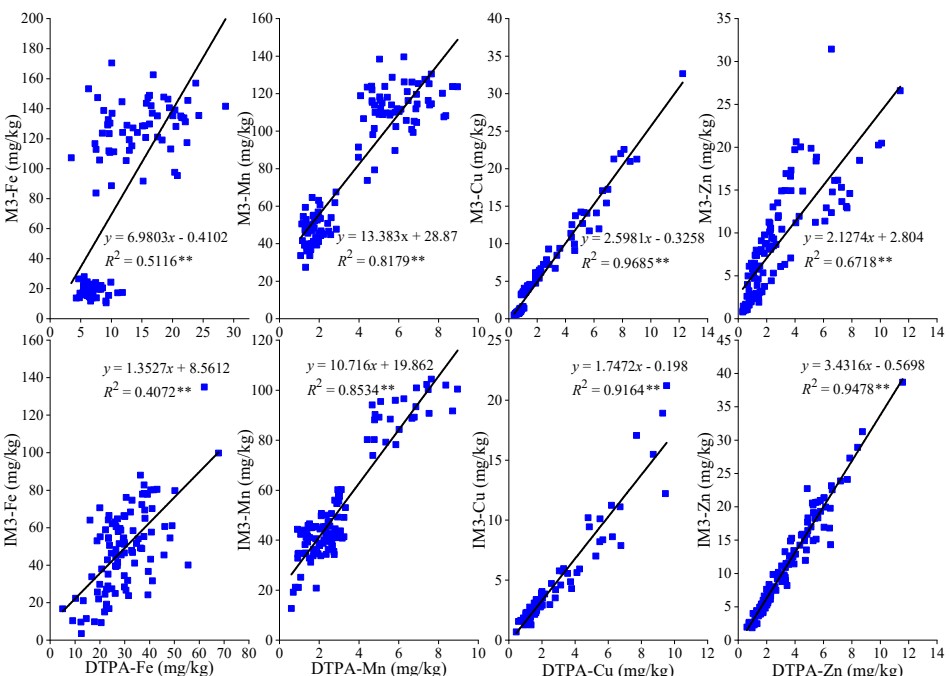

**Figure 3.** The correlations of the contents of soil available Fe, Mn, Cu and Zn determined by three test methods (** $p$ < 0.01).

### 3.2. Testing Precision Analysis of Three Test Methods

The testing precision, an important index to evaluate the feasibility of test method, could be characterized by the variation coefficient of the repeated results from the same samples under the same condition. The variation coefficients of available P, K, Fe and Cu determined by M3 and IM3 methods were all lower than those determined by CT method (Figure 4). The variation coefficients of available Mn and Zn determined by IM3 were 3.67% and 6.43%, respectively, which were slightly higher than those determined by CT method (2.72% and 5.29%), but lower than those determined by M3 method. This indicated that the testing precision of IM3 method could meet the requirements of soil test and formula fertilization technology.

### 3.3. Testing Time Analysis of Three Test Methods

On the premise that the testing precision can meet the testing requirements, the testing efficiency is another important index to be considered. The results in Table 3 indicated that the testing time varied greatly among the three test methods. For the CT method, the extracting liquid of available P, available K, and available Fe, Mn, Cu and Zn were obtained by extracting soil sample with $NaHCO_3$, $NH_4OAc$, and DTPA, separately. In addition, the other several steps including soil weighing, extractant adding, oscillating and filtering were operated independently. The total time needed for the six elements was 21.5 minutes/piece. The M3 and IM3 methods could simultaneously extract six elements, and only require 6.5 and 6.3 minutes/piece, respectively. Overall, in comparison with CT method and M3 method, IM3 method significantly improved the testing efficiency with the testing time being saved by 70.70% and 3.08%.

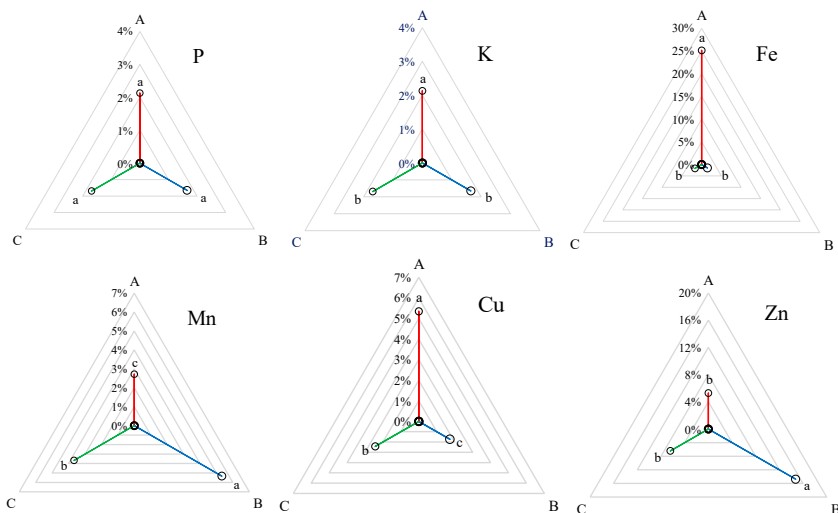

**Figure 4.** The variation coefficient of soil available in P, K, Fe, Mn, Cu and Zn determined by three test methods. Note: A, B and C indicated CT method, M3 method and IM3 method, respectively. The different letters of a, b and c indicated the significance at $p < 0.05$.

**Table 3.** The comparison of testing time among three test methods.

| Step | CT Method (min/piece) | | | M3 Method (min/piece) | IM3 Method (min/piece) |
|---|---|---|---|---|---|
| | **P** | **K** | **Fe, Mn, Cu, Zn** | **P, K, Fe, Mn, Cu, Zn** | **P, K, Fe, Mn, Cu, Zn** |
| Extractant preparation | 1.5 | 1 | 2 | 3 | 2.8 |
| Soil weighing | 1 | 1 | 1 | 1 | 1 |
| Extractant adding | 1 | 1 | 1 | 1 | 1 |
| Oscillating and filtering | 2.5 | 2.5 | 6 | 1.5 | 1.5 |
| Total | | 21.5 | | 6.5 | 6.3 |

### 3.4. Testing Cost Analysis of Three Test Methods

The testing cost is another important factor to be considered when determining the content of available nutrients in large amount of samples. As can be seen from Table 4, the reagent cost of CT method was 1.48 yuan/piece for the determination of available P, 0.27 yuan/piece for available K, and 0.20 yuan/piece for available Fe, Mn, Cu and Zn, respectively. Meanwhile, three pieces of quantitative filter paper were needed during filtration and the total cost was 0.99 yuan/piece. For M3 and IM3 methods, the reagent cost of six elements determination was 1.21 yuan/piece and 0.81 yuan/piece, respectively. Only one-time filtration was needed and so one piece of filter paper was required for obtaining extract liquid of six elements. Therefore, the cost of filter paper was only 0.33 yuan/piece. The total costs of extracting liquid by CT, M3, and IM3 methods were 2.94 yuan/piece, 1.54 yuan/piece and 1.14 yuan/piece, respectively. The significant cost advantage was exhibited in M3 method and IM3 method compared with CT method. The improved IM3 method had the lowest testing cost, which was lower by 61.2% and 26.0% than the CT and M3 methods, respectively.

**Table 4.** The comparison of testing price among three test methods.

| Methods (yuan/piece) | Reagent | | | Filter Paper | | | Total |
|---|---|---|---|---|---|---|---|
| | **P** | **K** | **Fe, Mn, Cu, Zn** | **P** | **K** | **Fe, Mn, Cu, Zn** | |
| CT | 1.48 | 0.27 | 0.2 | 0.33 | 0.33 | 0.33 | 2.94 |
| M3 | | | 1.21 | | | 0.33 | 1.54 |
| IM3 | | | 0.81 | | | 0.33 | 1.14 |

## 4. Discussion

### 4.1. Analysis of Extracting Principle of Improved IM3 Method

M3 extractant (pH 2.5 ± 1) was composed of 0.2 mol/L HOAc, 0.25 mol/L $NH_4NO_3$, 0.015 mol/L $NH_4F$, 0.013 mol/L $HNO_3$, and 0.001 mol/L EDTA [20,21]. Fluoride ion is a strong extractant of Al-P with a strong complexation ability regarding aluminum. Likewise, it also has a certain complexing ability for iron ions. EDTA, as a metal chelating agent, can chelate iron, aluminum, and calcium from Fe-P, Al-P and Ca-P, so that the solid-phase phosphorus could be released into the extract solution [22,23]. Thus, EDTA is a good general extracting agent for determination of soil available P in various types of soil. The ammonium ions and sodium ions in the extracting agent have strong substitution ability to extract exchangeable potassium from the soil. Meanwhile, available Fe, Mn, Cu and Zn can be extracted from the soil due to the chelation and complexation of metal elements by EDTA and fluoride ions [24].

Based on analyzing the extraction principle of M3 method, the replacement of ammonium nitrate contained in extractant of M3 method, ammonium chloride in extractant of IM3 method can also provide ammonium ion and play the same substitution effect as ammonium nitrate. Meanwhile, this replacement had addressed some problems including the difficulty in purchasing it and the high risk in storage and usage. The results of this study also showed that the improved IM3 method could achieve a similar or even better extraction effect than M3 method when applied to the combined extraction of available P, available K, available Fe, Mn, Cu and Zn in calcareous soil (Figure 5). The pH of the extraction solution of IM3 was increased from 2.5 to 3.6 because of removing nitric acid from extractant of M3 method. This may be why the contents of components by IM3 method were lower than that by M3 method and were closer to the results by CT method. In addition, the greatly higher results obtained by M3 and IM3 methods were also possibly related to the lower pH (2.5 and 3.6) compared with the results by CT method (7.0–8.5). Zhang et al. (2018) also found the similar results [18]. These results indicated that the pH of the extractant should be the main contributor to affect the extractivity. The extractant with lower pH easily activated the insoluble compound in soil, which may not have been available for crops in short time and could not be extracted by CT method with high pH. Therefore, the optimum pH of combined extractant suited for calcareous soil should be paid more attention in the future.

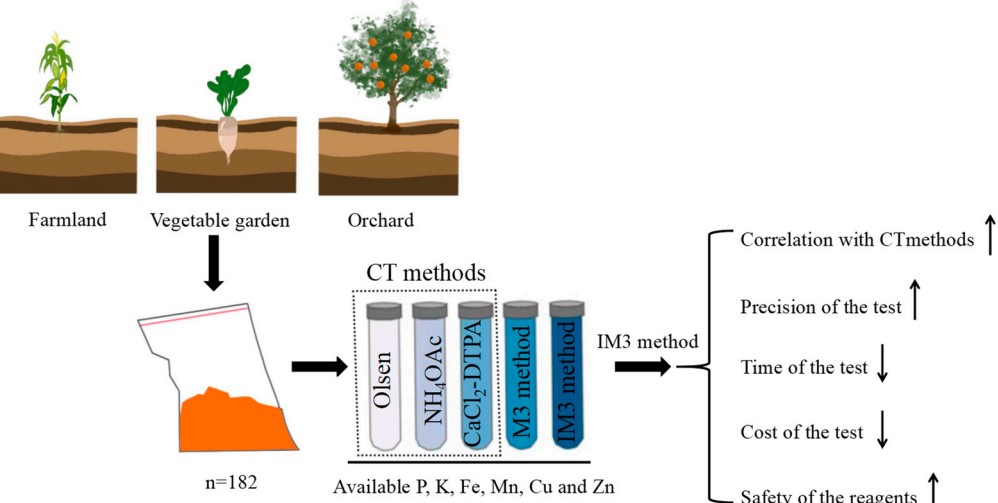

**Figure 5.** An illustration graph of the results of the three test methods.

### 4.2. Feasibility Analysis of IM3 Method

In this study, the contents of available P, available K and available Fe, Mn, Cu and Zn in calcareous soil determined by M3 and IM3 methods demonstrated extremely significant

linear correlation with the results determined by CT method. The correlation coefficients of available Zn and Mn between IM3 method and CT method were improved significantly compared with that between M3 method and CT method. The $R^2$ of available Zn increased from 0.6718 to 0.9478, and that of available Mn increased from 0.8179 to 0.8534. However, the correlation coefficients of available Fe were both lower between M3 method and CT method, and between IM3 method and CT method. The possible reason was that Fe in soil is mainly presented as soluble state, exchangeable state, chelated state, organic-bound state, oxide state, carbonate-bound state, mineral state, etc. [25]. DTPA, as an extractant with pH 7.0 in CT method, mainly extracts soluble Fe, exchangeable Fe and chelated Fe, while the combined extractant with low pH (2.5 and 3.6) in M3 and IM3 method could extract more than other forms of Fe, especially regarding indissolvable states [26]. Elrashidi et al. [27] also found a low correlation coefficient (only 0.59) of available Mn content in calcareous soil between M3 method and CT method. This similar reason may be explained for the unsatisfactory correlation between M3-Mn, IM3-Mn and DTPA-Mn (CT method) [28].

The testing precision is the most important index to evaluate the feasibility of test method. Meanwhile, the abundance and deficiency degrees of soil available nutrients are important parameters for fertilizer formula design and rational fertilization. Therefore, the determination results of soil available nutrients must be obtained as quickly as possible on the premise of ensuring the testing precision. In addition, the lower testing cost was essential. This study indicated that the testing precision of IM3 method could meet the requirement of popularizing soil testing and formula fertilization technology. The testing time for determining available P, K, Fe, Mn, Cu and Zn simultaneously by IM3 method was reduced by 70.7% and 3.1%, and the testing cost of IM3 method was reduced by 61.2% and 26% compared with those by CT and M3 methods, separately. In view of this, IM3 method is an ideal rapid measurement method for realizing simultaneous extraction of various soil nutrients in calcareous soil. However, further studies are needed to determine whether IM3 method is suitable for other neutral and acidic soils. Furthermore, field calibration studies on checking the correlation between the test values and crop responses were necessary for more accurately evaluating the feasibility of test method.

## 5. Conclusions

The results of available P, available K, and available Fe, Mn, Cu and Zn in calcareous soil determined by improved IM3 method showed a significant correlation with the results determined by CT method. On the premise of ensuring the testing precision, the improved IM3 method could not only save the testing time and reduce the testing cost, but could also improve the testing safety, operability and convenience due to avoiding the usage of ammonium nitrate and nitric acid.

According to the fitting equations, the nutrient abundance and deficiency indexes for calcareous soil suited for IM3 method could be established. The related results would provide the technical support for the design of fertilizer formula and the popularization of scientific fertilization technology.

**Author Contributions:** Methodology, Z.S.; software, C.L. and C.X.; formal analysis, S.W. and Z.S.; investigation, C.L. and M.S.; data curation, H.X. and L.Z.; writing—original draft preparation, S.W., M.S. and C.L.; writing—review and editing, D.L. and Z.S.; supervision, Z.S. and W.M. All authors have read and agreed to the published version of the manuscript.

**Funding:** This work was financially supported by National Key Research and Development Program of China (2021YFD1901004; 2016YFD0200403).

**Institutional Review Board Statement:** Not applicable.

**Informed Consent Statement:** Not applicable.

**Data Availability Statement:** Not applicable.

**Conflicts of Interest:** The authors declare no conflict of interest.

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
