# Peer review of "Improving the Extraction Process of Mehlich 3 Method for Calcareous Soil Nutrients"

_agronomy, doi:10.3390/agronomy12112907_

Round 1

Reviewer 1 Report

Authors improved the Mehlich3 method for rapidly analyzing soil nutrients. Compared with the Mehlich3 method, the improved Mehlich3 method avoided the use of hazardous ammonium nitrate and nitric acid. Compared with the conventional method, the improved Mehlich3 method reduced the testing varitation and improved the testing precision of several nutrients. The improved Mehlich3 method is a useful tool to rapidly analyzing soil nutrients, which is of great significance to soil nutrient mangement and fertilization mangement in agriculture. However, there are much language and word mistakes in this manuscript, it is suggested to ask help from a English native speaker to polish this manuscript. Above all, this paper needs major revision before acceptance.

1 Line 14-26, the language of this part is redundant and obscure, and it exits some basic format errors, please rehearse it carefully. The abstract needs to be more concise and explicit.

2 Line 45, Keywords, I would recommend to place the “Calcareous soil” to the end and replace the “Nutrient repiad test method” by “improved testing method for soil nutrients”.  

3 The introducution is not hypothesis driven, it is lack of theoretical support to some extent, the dicsussion has the same problem, thus the dicussion is short of logic. For example, In Line 221-Line 225, the unuse of ammonium nitrate will lead to what kind of change on the extract ions ? What’s the difference influence of the ammonium nitrate and amonium chloride on the soil extracts ?

4 Line 50, “the increase of agricultural production cost” and “the decline of economic benefit” seems to be with the same meaning.

5 Line 54, testing method is not a solution to the fertilizer utilization efficiency problem, it is recommended that “soil testing and formula fertilization technology is an effective way to uncover the soil nutrient contents, which is an important contributor to realize the high nutrient utilization efficiency”.

6 Line 58, there is needed a blank between the bracket and the words, find and revise this mistake thoughout the paper.

7 Line 68, this sentence is hard to understand

8 Line 95, Rehearse this sentence, carefully understand and use the "development".

9 Some other words or sentences need to be revised or are hard to understand is marked in Line 109, Line 113-135, Line 124, Line 136, Line 148, Line 151-154, Line180, Line 190, Line 211, Line 227, Line 235, Line 244, Line 272-286.

Author Response

Dear Reviewer:                           

We thank the reviewer for their useful comments. We have revised our manuscript carefully. The responses to reviewer’s comments are provided as below.

1 Line 14-26, the language of this part is redundant and obscure, and it exits some basic format errors, please rehearse it carefully. The abstract needs to be more concise and explicit.

Response: Thanks for your comment. We have improved our English expression and rewritten the abstract using clear and concise language. Please see line 14-38.

2 Line 45, Keywords, I would recommend to place the “Calcareous soil” to the end and replace the “Nutrient repiad test method” by “improved testing method for soil nutrients”.

Response: Thanks for your comment. We have added the “Calcareous soil” to the end of keywords and replaced the “Nutrient rapid test method” by “improved testing method for soil nutrients”. Please see line 39-40.

3 The introduction is not hypothesis driven, it is lack of theoretical support to some extent, the discussion has the same problem, thus the discussion is short of logic. For example, In Line 221-Line 225, the unuse of ammonium nitrate will lead to what kind of change on the extract ions ? What’s the difference influence of the ammonium nitrate and amonium chloride on the soil extracts ?

Response: Thanks for your valuable comment. We have provided theoretical references and enhanced logical expression in the introduction and discussion. Also, the relative explanation had been supplemented. Please see line 70-77 and line 224-225.

4 Line 50, “the increase of agricultural production cost” and “the decline of economic benefit” seems to be with the same meaning.

Response: Thanks for your comment. We have deleted “the decline of economic benefit”. Please see line 45.

5 Line 54, testing method is not a solution to the fertilizer utilization efficiency problem, it is recommended that “soil testing and formula fertilization technology is an effective way to uncover the soil nutrient contents, which is an important contributor to realize the high nutrient utilization efficiency”.

Response: Thanks for your comment. We have changed this expression. Please see line 50-51.

6 Line 58, there is needed a blank between the bracket and the words, find and revise this mistake though out the paper.

Response: Thanks for your comment. We have corrected all such errors in the manuscript.

7 Line 68, this sentence is hard to understand

Response: Thanks for your comment. We have rewritten this sentence. Please see line 60-61.

8 Line 95, Rehearse this sentence, carefully understand and use the "development".

Response: Thanks for your comment. We have rewritten this sentence. Please see line 98.

9 Some other words or sentences need to be revised or are hard to understand is marked in Line 109, Line 113-135, Line 124, Line 136, Line 148, Line 151-154, Line180, Line 190, Line 211, Line 227, Line 235, Line 244, Line 272-286.

Response: Thanks for your comment. We have revised and polished the paper.

Reviewer 2 Report

The manuscript agronomy-1973999 "Improvement of rapid measurement method for calcareous soil 2 nutrients" is devoted to the actual problem of soil analysis - the improvement of extraction methods.

The article is very interesting, but there are several serious remarks that the authors should eliminate:

1) The authors worked in fairly wide ranges of concentrations for each of the components being determined. However, they did not discuss how the concentration of the analyte affects its extractivity.

2) Unfortunately, the authors did not use standard samples in their experiment (i.e., samples with precisely specified concentrations of one or another component). This does not allow us to assess how the proposed solvents completely extract the components to be determined.

3) According to Figure 3, the M3 method gives overestimated concentrations of components from 2.12 to 13.38 times compared to the DTPA method, and the IM3 method underestimates. The authors do not discuss this! What could be the reason for this and which method gives a more complete extraction?

Small remarks:

1) The title of the article must be changed. Since the article is about improving the extraction process, and not the determination of nutrients in soils.

3) In section 2.1. Soil sampling and characterization authors should add soil names according to WRB (2015) soil classification.

4) It is necessary to explain why the authors took soil samples from a depth of 0-20 cm, and not along soil horizons?

5) In Table 2 "Soil:Water" should be replaced by "Soil:Solution".

6) Lines 123-128. The purity of the reagents and the manufacturer must be indicated. Because this greatly affects the cost of reagents.

Author Response

Dear Reviewer:                           

We thank the reviewer for their useful comments. We have revised our manuscript carefully. The responses to reviewer’s comments are provided as below.

1 Unfortunately, the authors did not use standard samples in their experiment (i.e., samples with precisely specified concentrations of one or another component). This does not allow us to assess how the proposed solvents completely extract the components to be determined.

Response: Thanks for your comment. In this experiment, we have used standard samples to ensure the accuracy of the test results. However, due to our negligence, this part of the explanation is not reflected in the paper. We have added it to the materials and methods. Please see line 85-88.

2 The authors worked in fairly wide ranges of concentrations for each of the components being determined. However, they did not discuss how the concentration of the analyte affects its extractivity. According to Figure 3, the M3 method gives overestimated concentrations of components from 2.12 to 13.38 times compared to the DTPA method, and the IM3 method underestimates. The authors do not discuss this! What could be the reason for this and which method gives a more complete extraction?

Response: Thanks for your professional question. The higher results by M3 method than that by CT method was also found by Zhang et al (2018). M3 had the highest extraction value, but not mean the more complete extraction. The reason is that the extractant with lower pH in M3 method easily activated the insoluble compound in soil. The contents of components by IM3 method was lower than that by M3 method and was more close to the results by CT method. The related reasons of higher results and affecting extractivity had been supplemented in discussion. Please see line 235-246.

(Zhang, C.P.; Niu, D.C.; Ren,Y.T.; Fu, H. Extractability of nutrients using Mehlich 3 and ammonium bicarbonate-DTPA methods for selected grassland soils of China. Plant, Soil and Environment, 2018, 64(No. 9), 448-454.)

Small remarks:

1 The title of the article must be changed. Since the article is about improving the extraction process, and not the determination of nutrients in soils.

Response: Thanks for your comment. We have revised it according to your suggestion. Please see line 2-3.

2 In section 2.1. Soil sampling and characterization authors should add soil names according to WRB (2015) soil classification.

Response: Thanks for your comment. We have revised it according to your suggestion. Please see line 89-91.

3 It is necessary to explain why the authors took soil samples from a depth of 0-20 cm, and not along soil horizons?

Response: Thanks for your question. The abundance and deficiency degrees of available nutrients in 0-20cm soil layer concentrated with roots are the most important parameters for fertilizer formula design and rational fertilization.

4 In Table 2 "Soil:Water" should be replaced by "Soil:Solution".

Response: Thanks for your comment. We have replaced it in this paper. Please see line 108.

5 Lines 123-128. The purity of the reagents and the manufacturer must be indicated. Because this greatly affects the cost of reagents.

Response: Thanks for your comment. We have added relevant information in this paper. Please see line 128-129 and line 133-134.

Reviewer 3 Report

 Improvement of rapid measurement method for calcareous soil nutrients

 Dear authors,

Herewith are suggestions for improving the manuscript. Please, go through the manuscript and improve it as per the suggestion. 

Line 2: . . .  rapid soil measurement… your title must be too specific and understandable to the use before going deep down to the content of the article.

Line 19: All abbreviations must be spelled out at their first appearance, e.g., ASI and IM3.

Line: 49 “unscientific fertilization” . . . can be explained as blanket recommendations.

Line 65: “showed a good correlation” . . . use professional terms good = significant? Can better express in the context of the statement.

Line 87: All samples were . . .

Line 93: Improvement of M3 test methos can be represented by Improved of M3 method.

Line 100 – 105: Table 2. Punction used to separate the extractants must be the appropriate comma ( , ).

Line 110:  Make the instrument used to analyze the extractant. ICP-OES (manufacturing company, state, country?

Line 123: mention the type and name filter paper used (manufacturing company, state, country).

These suggestions are also included as 'insert text at the cursor’ in the pdf file of the manuscript.

With best,

The reviewer

Author Response

Dear Reviewer:                           

We thank the reviewer for their useful comments. We have revised our manuscript carefully. The responses to reviewer’s comments are provided as below.

1 Line 2:. . .rapid soil measurement… your title must be too specific and understandable to the use before going deep down to the content of the article.

Response: Thanks for your comment. We have revised the title of this paper. Please see line 2-3.

2 Line 19: All abbreviations must be spelled out at their first appearance, e.g., ASI and IM3.

Response: Thanks for your comment. We have supplemented it in this paper. Please see line 19 and 55-56.

3 Line: 49 “unscientific fertilization” . . . can be explained as blanket recommendations.

Response: Thanks for your comment. We have changed the word. Please see line 44.

4 Line 65: “showed a good correlation” . . . use professional terms good = significant? Can better express in the context of the statement.

Response: Thanks for your comment. We have revised it in this paper. Please see line 58.

5 Line 87: All samples were . . .

Response: Thanks for your comment. We have revised the expression in this paper. Please see line 83.

6 Line 93: Improvement of M3 test methos can be represented by Improved of M3 method.

Response: Thanks for your comment. We have revised it in this paper. Please see line 97.

7 Line 100 – 105: Table 2. Punction used to separate the extractants must be the appropriate comma ( , ).

Response: Thanks for your comment. We have revised it in this paper. Please see line 108-109.

8 Line 110: Make the instrument used to analyze the extractant. ICP-OES (manufacturing company, state, country?

Response: Thanks for your comment. We have added related information in the paper. Please see line 103-107.

9 Line 123: mention the type and name filter paper used (manufacturing company, state, country).

These suggestions are also included as 'insert text at the cursor’ in the pdf file of the manuscript.

Response: Thanks for your comment. We have added the related information in the paper. Please see line 127-129.

Round 2

Reviewer 2 Report

The authors have corrected all comments. The manuscript may be published. One small correction: line 134 "fluvisol soil" please change to "Fluvisol".